# The Diagnostic Potential of Amyloidogenic Proteins

**DOI:** 10.3390/ijms22084128

**Published:** 2021-04-16

**Authors:** Yiyun Jin, Devkee Mahesh Vadukul, Dimitra Gialama, Ying Ge, Rebecca Thrush, Joe Thomas White, Francesco Antonio Aprile

**Affiliations:** Department of Chemistry, Molecular Sciences Research Hub, Imperial College London, London W12 0BZ, UK; y.jin@imperial.ac.uk (Y.J.); d.vadukul@imperial.ac.uk (D.M.V.); dimitra.gialama07@imperial.ac.uk (D.G.); y.ge19@imperial.ac.uk (Y.G.); r.thrush20@imperial.ac.uk (R.T.); joe.white17@imperial.ac.uk (J.T.W.)

**Keywords:** neurodegenerative diseases, biomarker, amyloid, oligomer, post-translational modification

## Abstract

Neurodegenerative disorders are a highly prevalent class of diseases, whose pathological mechanisms start before the appearance of any clear symptoms. This fact has prompted scientists to search for biomarkers that could aid early treatment. These currently incurable pathologies share the presence of aberrant aggregates called amyloids in the nervous system, which are composed of specific proteins. In this review, we discuss how these proteins, their conformations and modifications could be exploited as biomarkers for diagnostic purposes. We focus on proteins that are associated with the most prevalent neurodegenerative disorders, including Alzheimer’s and Parkinson’s diseases, amyotrophic lateral sclerosis, and frontotemporal dementia. We also describe current challenges in detection, the most recent techniques with diagnostic potentials and possible future developments in diagnosis.

## 1. The Need for Novel Diagnostic Approaches for Neurodegeneration

Neurodegenerative diseases are fatal and incurable disorders, characterized by the progressive loss of neurons in specific regions of the nervous system. They are a highly heterogeneous group of pathologies, which include Alzheimer’s disease (AD), Parkinson’s disease (PD), frontotemporal dementia (FTD) and amyotrophic lateral sclerosis (ALS). Currently, worldwide, more than 50 million people suffer from various forms of neurodegeneration [1].

The clinical course of neurodegenerative diseases usually spans several years and leads to progressive deficits in memory, cognition, and movement to different extents depending on the specific pathology [2]. Existing drug treatments focus on the relief of these symptoms [3]. Furthermore, neuropsychological assessment is still considered crucial in the diagnosis neurodegeneration associated with dementia [4]. However, it is apparent that key molecular mechanisms of disease occur before the appearance of any significant symptoms. The development of accurate diagnostic approaches would facilitate timely therapeutic interventions to restore neuronal physiology before irreversible damage occurs. It would also promote the establishment of new therapeutics, and the revaluation of current ones which could be more effective if administrated at earlier stages. Despite the urgent need for diagnostic approaches for neurodegeneration, their development is still a daunting challenge, due to the limited accessibility of the brain for physical examination and the complexity of clinical tests based on cognitive abilities [2].

Recent technological advances have enabled the characterization of novel pathways, biomolecules, and structures in the nervous system and other regions of the body that could be used as disease markers of neurodegeneration [5,6]. Although the different neurodegenerative diseases have some distinct phenotypes, they also share some key molecular features. In particular, in many of these disorders, specific proteins and peptides, which would normally be soluble, undergo a self-assembly process which leads to the formation of large fibrillar aggregates, called amyloids [7,8]. This process also involves the generation of smaller oligomeric intermediates, which are highly toxic and currently regarded as major players in the disease mechanisms [8]. Amyloid aggregation is also linked to other aggregation processes, such as the formation of condensates [9]. However, these will not be the focus of the present review.

Here, we describe some of the most known amyloidogenic proteins and their diagnostic relevance. We focus on amyloid-beta (Aβ) and tau for AD, α-synuclein (α-syn) for PD, fused in sarcoma (FUS) and the TAR DNA-binding protein 43 (TDP-43) for ALS and FTD. We also discuss state-of-the-art advancements in detection approaches to monitor the aggregation of these proteins.

## 2. Amyloid Aggregation as Potential Source of Biomarkers

Amyloids are insoluble fibrillar aggregates enriched in a cross-β structure, and their formation has been extensively characterized in vitro [7,8]. Amyloid aggregation consists of a complex network of nucleation events. Initially, soluble monomeric proteins interact and form oligomers by primary nucleation. Primary nucleation can also be triggered by the presence of other biomolecules, such as other proteins [10], nucleic acids [11,12] and membranes [13]. Oligomers then convert into higher-order aggregates and, finally, into amyloid fibrils. Once a critical concentration of fibrils has formed, the surface of these fibrils catalyzes the formation of additional oligomers by secondary nucleation [7,8]. Fibrils can also elongate by the addition of monomers at their ends, and undergo fragmentation [7,8].

Amyloid fibrils are extremely stable protein species due to their rich cross β-sheet content [8,14,15,16]. On the contrary, oligomers rapidly convert into higher order amyloid aggregates (Figure 1). The transient nature of oligomers makes them difficult to isolate and characterize at the structural level. Only recently, thanks to new approaches such as single molecule fluorescence and electron microscopy [14,17,18], has it been shown that oligomers are highly heterogeneous in their physiochemical properties and structures, with a varying secondary structure content [16,19,20].

Oligomers are toxic by a multitude of mechanisms including aberrant hydrophobic interactions. It has been shown that, in tissue and isolated cells, oligomers can affect membrane permeability, ion homeostasis, and induce oxidative stress [16,21,22,23]. Oligomer-induced free radicals can then trigger protein misfolding, mitochondrial dysfunction and eventually apoptosis [24]. Oligomers of Aβ and α-syn have also been associated with neuroinflammation [22] and synapsis loss [25,26].

Several factors can affect the formation of amyloid fibrils and oligomers. These include genetic mutations, cellular stress, and the presence or absence of specific biomolecules. Furthermore, amyloids are extensively post-translationally modified in vivo [27,28,29] and post-translational modifications (PTMs) significantly alter the formation and the toxicity of amyloid fibrils in vitro [27,28,29,30,31].

## 3. Amyloidogenic Proteins Involved in Neurodegeneration

In the context of neurodegeneration, disease markers can be divided into neuropsychological, neuroimaging, genetic and biochemical markers [2]. In particular, biochemical markers (or biomarkers) are measurable molecules in our body (e.g., proteins, nucleic acids, metabolites), which report the stage of a disease [32]. Amyloidogenic proteins are promising biomarkers, as they inform on the biochemical profile of the nervous system dysfunction [2]. Below, we highlight relevant amyloidogenic proteins and their pathological modifications which can serve as biomarkers for neurodegenerative conditions.

### 3.1. Aβ and Tau in AD

AD is the most prevalent form of dementia. The characteristic lesions in AD brains are extracellular senile plaques composed of amyloid aggregates of Aβ and intracellular neurofibrillary tangles (NFTs) formed by paired helical amyloid filaments (PHFs) of hyperphosphorylated tau (p-tau) protein [2,27].

Aβ is a short peptide generated by the cleavage of a larger transmembrane precursor, called the amyloid precursor protein (APP), by the sequential cleavage of the β- and γ-secretases and released into the extracellular space [2,33]. This process can generate Aβ isoforms of various lengths (Table 1), which have various degrees of toxicity in the context of AD [34]. The most common Aβ isoforms are the 40- and 42-residue long ones, generally referred to as Aβ40 and Aβ42, respectively. Aβ40 is the most abundant variant in the plaques (~80% to 90%) and is also present in healthy peoples’ brains. Aβ42 has a much higher propensity to aggregate, and an increase in Aβ42/Aβ40 ratios is associated with AD and other forms of dementia [30,34] (Table 2). Besides cleavage, genetic mutations (A692G, E693Q in the APP gene [35,36], Table 2) and many other PTMs of Aβ have been associated with AD, including oxidation, phosphorylation, glycosylation and isomerization [30]. Studies show the occurrence of acetylation (e.g., Lys16 and Lys28), phosphorylation (e.g., Ser8 and Ser26), nitration (e.g., Tyr10), pyroglutamation (e.g., Glu3 and Glu11), isomerization (e.g., Asp1 and Asp7) and racemization (e.g., Asp1, Asp23 and Ser26) in the context of disease [30,37,38,39] (Table 1). It is worth noting that cognitive decline correlates more with soluble intermediate forms of Aβ rather than the degree of amyloid deposits [16].

Tau is a major microtubule-associated protein that stabilizes the microtubules in neurons [2,40]. In human brains, tau exists as six different isoforms that carry either three or four microtubule-binding repeats (R). These isoforms are called 3R and 4R, respectively. It has been found that the presence of either 3R or 4R or both 3R and 4R amyloids is disease specific. As an example, in AD, ALS, FTD and Parkinsonism, both 3R and 4R amyloids are present, while in corticobasal degenerations and Pick’s disease only 4R and 3R amyloids, respectively, are found [41,42]. Tau undergoes PTMs, particularly phosphorylation [40]. Pathological hyperphosphorylation reduces tau affinity for microtubules and causes its detachment from microtubules, resulting in the formation of PHFs and NFTs [2]. To date, 85 potential phosphorylation sites of tau have been identified [40]. Furthermore, molecular and cellular studies revealed that acetylation (e.g., Lys174, Lys274 and Lys280), oxidation (e.g., Cys322), nitration (e.g., Tyr29), glycation (e.g., Lys87, Lys132 and Lys150), truncation (e.g., at Asp13 and Asp421 and Glu391) and ubiquitination (e.g., Lys48 and Lys63) also affect tau aggregation [27] (Table 1).

### 3.2. α-Syn in PD

Unlike AD, PD primarily affects the motor system, causing tremors, rigidity, bradykinesia and postural instability [2]. The pathological hallmark of PD is the occurrence of cytoplasmic amyloid inclusions, known as Lewy bodies (LBs) and Lewy neurites (LNs). LBs and LNs are comprised of amyloid aggregates, whose main component is α-syn [43].

α-Syn contains 140 residues with a positively charged N-terminal region, an aggregation-prone *non-amyloid-β component* (NAC) central region and a negatively charged C-terminal region [44,45]. Duplications or triplications on the α-syn chromosome region (4q21-23) and mutations including A53T, G51D, H50Q, E46K, and A30P in the α-syn sequence, are associated with early-onset PD [46,47,48] (Table 2).

α-Syn belongs to a protein family which also includes β- and γ-synucleins with 55%–62% similarity. *β*-Synuclein has a reduced propensity to aggregate and has been discovered to suppress the aggregation of α-syn as a natural inhibitor while oxidized γ-synuclein can initiate α-syn aggregation [49,50]. Several PTMs are known to affect aggregation of α-syn, are associated with PD [28] and hold diagnostic potential [51] (Table 1). These include N-terminal acetylation, several truncations at the N-terminus (e.g., α-syn7-140, 14-140, 40-140, and 72–140 found in vitro, 5–140 and 68–140 found in vivo and several in both) and C-terminus (e.g., α-syn1-115, 1-119, 1-122, 1-124, 1-125, 1-129, 1-133, and 1-135), phosphorylation of Ser87 and Ser129, oxidation of Met1, Met5, Met116 and Met127, sumolyation of Lys96 and Lys102, nitration of Tyr39, Tyr125 and Tyr133, and ubiquitination of Lys6, Lys10, Lys12 Lys21, Lys23, Lys43 and Lys96 [28,52,53] (Table 1).

### 3.3. TDP-43 and FUS in ALS and FTD

ALS and FTD are neurodegenerative diseases with overlapping mechanisms. ALS affects upper and lower neurons, causing loss of muscle control. FTD is a form of dementia linked to the degeneration of the frontal and anterior temporal lobes [54]. Around 97% of ALS and 45% of FTD cases are associated with the presence of inclusions of aggregates of ubiquinated, hyperphosphorylated and C-terminally truncated TDP-43 in the cytoplasm of neurons and glial cells [29].

TDP-43 is a 414 residue-long ribonucleoprotein able to form amyloid-like aggregates in vitro and condensates (i.e., stress granules) [9] under pathological conditions. It is composed of an N-terminal tract with a nuclear localization signal, two RNA recognition motifs, a nuclear export signal, and a disordered C-terminal region [29]. All these regions have been reported to play a critical role in the aggregation of the protein [55,56,57]. Several TDP-43 mutations have been identified in both sporadic and familial cases of ALS and FTD, including G294A, Q331K, M337V [58] and K181E [59] (Table 2). As with Aβ and α-syn, PTMs of TDP-43 also play a key role in the aggregation of the protein and disease progression. It is worth noting that the truncated 25 kDa and 35 kDa C-terminal fragments are commonly found in pathological aggregates in ALS patients [29,31,54]. Ubiquitination is also a typical modification of TDP-43 inclusions [31]. Finally, aberrant phosphorylation, acetylation, and oxidation of TDP-43 is often associated with the mislocalization and aberrant aggregation of the protein [29] (Table 1).

ALS and FTD are also associated with another RNA/DNA-binding protein, FUS. FUS is a 526 residue-long protein made by an N-terminal transcriptional activation domain and C-terminal domain, which interacts with transcriptional factors and also includes a nuclear localization signal [60,61]. Both domains contain low complexity regions and play a role in the formation of condensates and hydrogels [62]. More than 50 mutations (e.g., R521C, R521H [63]) of FUS are reported in ALS/FTD cases. Unlike TDP-43, FUS is generally found as a full-length protein in the aggregates [61]. Phosphorylation occurs in the prion-like domains of FUS and has been shown to affect its phase separation and aggregation pattern while mutations and PTMs (mainly methylation and phosphorylation) on its C-terminal domain are found to regulate its nuclear/cytoplasm localization [61,64]. Despite the clear pathological role of FUS, ALS/FTD phenotypes are less frequently associated with FUS than with TDP-43 dysfunctions [61,65]. Thus, the role of FUS as biomarker remains to be determined.

## 4. Diagnostic Potential of Genetic, Structural and Chemical Features of Amyloidogenic Proteins

Several detection approaches have been used to quantify amyloidogenic proteins in biological samples and to determine the link between these proteins and neurodegenerative diseases. Some of these strategies aim at quantifying the changes in the expression levels/concentration of amyloidogenic proteins regardless of their conformation or modification (Table 2). Some other approaches, instead, focus on probing specific structural (e.g., aggregated states) or chemical (i.e., PTM) properties of amyloidogenic proteins (Table 1).

All these approaches analyze different regions of the body. Techniques, such as positron emission tomography (PET), are able to probe proteins directly within the central nervous system (CNS), for example, in the brain [2,44,66]. However, amyloidogenic proteins can be also detected in other more accessible regions of the body. In this section, we discuss findings which have been obtained from the analysis of both brain tissues and accessible body fluids, mainly by means of immunoassays, such as immunoblotting and enzyme-linked immunosorbent assays (ELISA). The fluids under consideration include the cerebrospinal fluid (CSF), which is in direct contact with the extracellular portion of the brain and, as such, is an optimal fluid for measurements of brain metabolism [44].

### 4.1. Levels of Amyloidogenic Proteins in Neurodegeneration

So far, the most common detection strategies probe specific amyloidogenic proteins, regardless of their conformations or PTMs, providing information on the total amount of amyloidogenic protein in disease conditions. These methods are particularly effective when the pathological mechanism depends on genetic mutations (e.g., when a genetic variant of a protein is specifically expressed in patients, Table 2), protein expression levels (e.g., when protein expression is significantly altered in patients) or mislocalization (e.g., when a protein accumulates in specific regions of the body in patients).

In the case of AD, initial experiments focused on measuring the CSF concentration of total Aβ. However, no significant differences between patients and controls were found [33]. In contrast, significant differences have been observed for total tau. Researchers found a marked increase of total tau in the CSF of AD patients compared to healthy controls [67]. Although there is a clear link between the concentration of tau in the CSF and AD, higher concentrations of tau are also observed in acute stroke, brain trauma and other forms of dementia [66,68], making disease-specific diagnosis challenging [44,66].

In the case of PD, it has been found that α-syn is present at slightly lower concentrations in the CSF of patients compared to healthy individuals in a small cohort studies [68,69,70]. This has been observed also in plasma and saliva. In larger studies, CSF α-syn has a significantly lower concentration than healthy controls [69]. This lower concentration of α-syn in accessible body fluids may be a consequence of the intracellular aggregation of the protein [69]. However, the use of α-syn alone is not sufficient as a single biomarker for PD diagnosis [69].

Due to its low concentration, TDP-43 is challenging to reliably measure in the blood [71,72]. However, in the CSF, it has been shown that TDP-43 concentration is higher in ALS patients (6.92 ± 3.71 ng/mL) than in healthy subjects (5.31 ± 0.94 ng/mL) and people affected by other conditions, for example, PD, multiple sclerosis, and Guillain-Barré syndrome [73]. Recent studies have suggested that TDP-43 present in the CSF could be a biomarker for the diagnosis of ALS, ALS/FTD but not for FTD alone [74,75]. In particular, it should be noted that proteins, including TDP-43, can diffuse from the blood into the CSF. Furthermore, TDP-43 is also abundant in other parts of the body besides the brain [54,76]. Thus, TDP-43 in the CSF may originate from other body regions and may not adequately represent the brain pathology in FTD. Currently, research efforts are focusing on monitoring TDP-43 contained in CSF exosomes, which are brain-derived and thus better represent the brain pathology [77]. Overall, so far, despite compelling evidence of the fact that protein expression and concentrations are altered in disease, probing total protein concentration alone has not yielded a widely adopted diagnostic approach.

**Table 1 ijms-22-04128-t001:** Main PTMs of amyloidogenic proteins with diagnostic potential.

Protein	PTMs	Major Modification Sites	Key Remarks	Refs.
Aβ	Cleavage	1-37, 1-38, 1-39, 1-40, 1-42, 1-43	Aβ 40 is the most abundant; Aβ 42 aggregates more rapidly.	[30,78,79]
	Phosphorylation	Ser8, Ser26	Increased abundance and stability of toxic aggregates.	[30,80,81]
	Acetylation	Lys16, Lys28	Altered aggregation behavior.	[38,39]
	Oxidation	Met35	Regulation of oxidative stress. Slower aggregation.	[30,82]
	Nitration	Tyr10	Enhanced aggregation and plaque formation.	[30,37]
	Isomerization	Asp1, Asp7	Higher aggregation propensity and resistance to degradation.	[30]
	Racemization	Asp1, Asp23, Ser26	Higher aggregation propensity.	[30]
	O-glycosylation	Tyr10	Found in short Aβ fragments in AD patients’ CSF. Aβ1-15 and Aβ1-17 are the most abundant fragments.	[30,83]
	Pyroglutamate formation	Glu3, Glu11	Increased oligomerization. It correlates with the extent of Aβ deposition.	[84,85]
Tau	Phosphorylation	Thr181, Thr199, Thr217, Thr231	Increased aggregation. It is a key event in the formation of NFT.	[27,86]
	Acetylation	Lys174, Lys274, Lys280	Regulation of tau function; promotion of p-tau aggregation.	[87]
	Oxidation	Cys322	Enhanced PHF assembly	[88]
	Nitration	Tyr29	Accumulation of oligomeric species.	[89]
	N-glycosylation	Under investigation. Putative sites: Asn167, Ans359, Asn410	Higher levels in AD patients’ brains. Promotion of tau hyperphosphorylation and PHF accumulation.	[83,90]
	O-glycosylation	Under investigation	Lower levels in AD patients’ brains. It prevents tau hyperphosphorylation and PHF formation and PHF accumulation.	[91]
	Ubiquitination	Lys48, Lys63	It has been proposed to contribute to the formation of the tangles.	[92,93,94]
α-Syn	Fragmentation	C-terminally truncated α-syn of around 10–15 kDa	Accelerated aggregation.	[28]
	Phosphorylation	Ser129	Enhanced aggregation and toxicity.	[95,96]
	Acetylation	N-terminal	Increased helical propensity, altered fibril polymorphism and decreased aggregation.	[97]
	Oxidation	Met1, Met5, Met116, Met127	Inhibition of fibrillation by stabilization of soluble oligomeric species.	[98]
	Nitration	Tyr39	Enhanced aggregation.	[99]
	O-glycosylation	Under investigation.	Inhibition of aggregation.	[100]
	Ubiquitination	Lys6, Lys12, Lys21, Lys43	Inhibition of aggregation.	[28]
		Lys10, Lys23	Faster fibril formation.	[28]
TDP-43	Fragmentation	C-terminally truncated TDP-43 at 25 kDa and 35 kDa fragments	Enhanced aggregation, altered RNA processing, and cellular redistribution.	[101,102]
	Phosphorylation	Ser409, Ser410	Enhanced aggregation and cellular mislocalization.	[29,75]
	Acetylation	Lys145, Lys192	Impaired RNA binding and mitochondrial function; enhanced aggregation of phosphorylated TDP-43.	[29]
	Oxidation	Cys3, Cys50, Cys173, Cys175	Enhanced oligomerization and self-association.	[29]
	Ubiquitination	Lys 48, Lys 63	Enhanced cytoplasmic accumulation to higher molecular weight aggregates.	[29]

### 4.2. PTMs Associated with Neurodegeneration

As previously mentioned, PTMs affect the aggregation propensity of amyloidogenic proteins and play a critical role in the pathological mechanism of neurodegenerative diseases [27,28]. Furthermore, PTMs are also found in biological samples from patients [27,28,29]. Aberrant PTMs may occur well before clinically observable symptoms. This observation makes PTMs potentially suitable to be used as biomarkers for early diagnosis (Table 1).

#### 4.2.1. Fragmentation and Cleavage

Protein cleavage is a PTM commonly associated with amyloid aggregation and has been extensively investigated both in vitro and in vivo. In AD, for example, several APP fragments have been linked to the disease, Aβ42 and Aβ40 being the most extensively investigated ones. However, other forms of Aβ also hold a diagnostic potential, including Aβ37, Aβ38, Aβ39, and Aβ43 [32].

It has been found that the CSF concentration of Aβ42 is lower in AD patients with respect to controls in approximately 50% of cases [78]. However, currently, the CSF concentration ratio Aβ42/Aβ40 is considered diagnostically more accurate, as it takes into account the physiological inter-individual protein expression fluctuations [78,103]. Besides Aβ40 and Aβ42, the concentration of Aβ43 is lower in the CSF of patients with early-onset AD, but also in patients with mild cognitive impairment and other forms of dementia [79]. On the contrary, Aβ38 is present at higher concentrations in AD patients and specifically discriminates AD from other forms of dementia [66]. Thus, the ratio Aβ42/Aβ38 may correlate with different types of dementia and be used in combined detection diagnostic approaches [103,104].

Despite α-syn and TDP-43 also being able to undergo fragmentation, which plays a role in their pathological mechanisms, very few studies had detected truncated α-syn and TDP-43 isoforms in body fluids and their roles as potential biomarkers remained unknown [28,54,105,106].

#### 4.2.2. Phosphorylation

In Aβ, the phosphorylation of Ser8 and Ser26 is believed to increase the abundance and stability of toxic aggregates. Enrichment of these modifications in oligomeric Aβ species, compared to monomers, have been shown using phosphor-specific mono- or polyclonal antibodies [80,81].

As a consequence of hyper-phosphorylation, tau dissociates from the microtubule network and aggregates. Therefore, tau phosphorylation is considered a key event in the formation of NFT [27]. In the recent years, several antibodies have been developed to specifically recognize p-tau and a variety of ELISA protocols have been developed for different phosphorylated epitopes. Recently, p-tau at Thr217 (p-tau217) was shown to be a robust plasma [86] and CSF [107] biomarker for AD as it distinguished AD from other neurodegenerative diseases including other tauopathies.

Similarly, α-syn is also phosphorylated at multiple residues. Noteworthy to mention is the variant of α-syn carrying the phosphorylation of Ser129 (p-α-syn129). This modification occurs by multiple kinases and affects the aggregation and the toxicity of α-syn both in vitro [95,96] and in vivo [108]. Mass spectrometry (MS) and antibody-based experiments have found that LBs isolated from PD patients’ brains are enriched in p-α-syn129. They have also shown that the concentration of p-α-syn129 in body fluids is higher in PD patients and correlates with the disease severity [109,110]. The quantification of p-α-syn129, along with total α-syn, in the CSF was suggested to aid the modeling of PD and potentially the differential diagnosis of PD from other Parkinsonisms [110,111].

Several antibodies that are specific for phosphorylated TDP-43 (e.g., at Ser409 and Ser410) have been developed. However, TDP-43 undergoes several other modifications, and probing only the phosphorylation state of the protein does not accurately describe the disease state. Detection approaches currently under development involve the combined use of different antibodies targeting different modifications and conformations of TDP-43 [75,77].

#### 4.2.3. Acetylation

Acetylation is another major PTM involved in AD. In fact, it has been reported that the overall acetylation levels of Aβ42- and tau- aggregates are significantly higher in AD hippocampus compared to healthy subjects [112]. Furthermore, several studies have shown that the acetylation of Lys16 and Lys28 of Aβ both alter the aggregation of the peptide in vitro [38]. In particular, the acetylation of Lys16 leads to the formation of amorphous aggregates instead of amyloids [38,39].

Tau undergoes acetylation at various residues. In particular, acetylation at Lys174, Lys274 and Lys280 have received the most attention as these were shown to also regulate the function of the protein [87]. Among these modifications, Lys280 was specifically found acetylated in transgenic mice and patients’ hippocampus tissues [87].

In the case of PD and dementia with LBs (DLB), it has been reported that brain tissues (temporal cortex and dorsolateral prefrontal cortex) from patients contain soluble and aggregated N-terminally acetylated α-syn [113]. N-terminal acetylation has been found to significantly alter the structure of monomeric α-syn in vitro. This PTM induces the helical folding of the N-terminus of α-syn [97], which has been suggested to promote the pathological association of the protein to membranes [114].

TDP-43 is another protein that undergoes acetylation. This PTM abrogates RNA binding and promotes the aggregation of the protein. In particular, acetylated TDP-43 at Lys145 could be detected in lesions in ALS but not FTD patients, suggesting that TDP-43 acetylation was linked to ALS pathogenesis and could be used for discriminating ALS from FTD, which is helpful as a biomarker if easily detectable in body fluids [29].

#### 4.2.4. Oxidation and Nitration

Oxidative stress has been described as a contributing factor in neurodegenerative diseases [115]. In AD, oxidation of Aβ at Met35 was detected *post-mortem* in the occipital cortex in AD patients [116]. However, this modification was shown to slow down the aggregation of the protein in vitro [30,82] and further investigations will be needed to clarify the exact role in the disease. Instead of targeting oxidized amyloidogenic proteins, current diagnostic approaches mainly focus on measuring oxidative stress biomarkers (e.g., oxidation products and specific metabolites) [115]. However, these biomarkers are not specific for neurodegeneration, as oxidative stress is also induced by inflammation and in the context of various pathologies, including cancer [115]. The development of antibodies directed against oxidized amyloidogenic proteins, such as tau and TDP-43, coupled with highly sensitive approaches may help diagnosis.

Nitration of Tyr residues has been found in several amyloidogenic proteins from patients’ samples (e.g., Tyr10 in Aβ, Tyr29 in tau, Tyr39 in α-syn) [28,30,37,89]. In particular, it has been found that nitrated α-syn levels in peripheral blood cells (monocytes and erythrocytes) are specifically higher in PD patients. Thus, this modification is currently regarded as a potential diagnostic biomarker, although larger scale studies are required [99,117]. Nitrosative stress molecules are currently assessed as potential biomarkers. However, also in this case, they are not specific for neurodegeneration and further investigation is needed [118].

#### 4.2.5. Glycosylation

Studies found short Aβ O-glycopeptides (Aβ1-15 to 20) in AD CSF samples and suggested that they could be useful biomarkers, and the glycopeptide profiles and concentrations need to be confirmed in much larger prospective studies [119] (Table 1).

Tau has also been shown to undergo N- and O-glycosylation. Interestingly, O-glycosylation plays a protective mechanism against hyperphosphorylation and PHF formation [91]. In fact, lower levels of O-glycosylated tau are found in AD brains compared to healthy controls [91]. In contrast, N-glycosylation was shown to facilitate tau hyperphosphorylation, thus promoting its accumulation in PHF. N-glycosylation has been found in human AD brains, but not in control brains, which could be a relevant modification for diagnosis [83]. In PD, O-linked glycosylated with N-acetylglucosamine of α-syn was also found in vivo [100].

Diagnostic approaches based on the detection of glycosylated amyloidogenic proteins are still largely unexplored and new techniques are under development. The assessment of glycosylation patterns in patients include N-glycome profiling using glycoblotting and MS [120,121].

#### 4.2.6. Ubiquitination

Ubiquitin labels misfolded proteins and targets them for degradation through the ubiquitin-proteasome system. It is believed that abnormal accumulation of amyloids might result from the alteration of the ubiquitin-proteasome pathway [122].

It has been found that ubiquitin is present at higher concentrations in AD patients’ cortical tissues compared to healthy controls [122,123]. Although early studies have shown that the majority of the ubiquitination occurs when tau is aggregated in the tangles, ubiquitin has also been found in soluble tau, for example, via Lys63 conjugation [92,93]. Notably, Lys48 and, particularly, Lys63-linked polyubiquitination, as well as monoubiquitin modifications were found to contribute to the biogenesis of the tangles. Further analyses could focus on the detection of Lys63 ubiquitinated tau to evaluate its diagnostic value [94].

α-Syn is found, mono-, di- and tri-ubiquitinated, in LBs from the cortical tissues of autopsied DLB patients’ brains [124]. While ubiquitination at Lys6, Lys12, Lys21, Lys43, and Lys96 has been shown to have an inhibitory effect on fibril formation, ubiquitination at Lys10 and Lys23 has the opposite effect and could have a pathological role. However, their presence in body fluids and connection with the PD pathology remain unknown [28].

Ubiquitinated TDP-43 is a major component of the aggregates in about 50% of FTD cases and of 97% of ALS cases, and appears to be downstream of phosphorylation. Nevertheless, the effects of ubiquitination on TDP-43 aggregation still need to be further clarified [29]. Currently, potential diagnostic approaches focus on measuring free nonconjugated ubiquitin rather than ubiquitinated proteins in the CSF [125]. It has been observed that the amount of ubiquitin in the CSF reflects the amount of protein aggregates in the brain and could serve as biomarker [125,126].

### 4.3. Amyloid Aggregates and Amyloid-Sensitive Probes

Currently, increasing research effort is focusing on generating biomolecular tools to target specific aggregate conformations (Figure 2). The rationale underlying this strategy consists of the fact that amyloidogenic proteins are generally poorly toxic when monomeric, while their toxicity significantly increases when they are aggregated [13]. Recent evidence shows that oligomers are particularly toxic and [13], thus, in order to monitor the progression and severity of neurodegenerative diseases, one could specifically quantify the concentration of the amyloids and/or of the oligomers (Table 2, Figure 3).

In this context, several probes for amyloids and amyloid-like aggregates have been developed, including small molecule fluorescent probes, conformation-selective antibodies and aptamers [127,128,129,130].

In particular, single-molecule studies using amyloid-sensitive fluorescent molecules were able to probe individual soluble aggregates present in body fluids of patients. To do so, small molecules, such as thioflavin T (ThT), which increase their fluorescence upon binding to the cross-β-sheet component of the amyloids, have been used. Using this approach, researchers managed to quantify soluble amyloid-like aggregates in AD and PD CSF samples [131,132]. Besides ThT, other molecules with similar fluorescent properties have been developed, including molecular rotors [133,134], such as thioflavin X [135].

In addition to these small molecule probes, high-specificity biomolecules have been developed, such as antibody fragments and aptamers able to probe oligomers of different amyloidogenic proteins [127,136,137]. Recently, super-resolution microscopy with amyloid-specific aptamer and DNA point accumulation in nanoscale topography was able to detect aggregated α-syn and Aβ in samples from PD patients without the need of a conjugated fluorophore [138].

Traditionally, antibodies have been generated using well-established techniques in vivo by immunization and hybridoma technologies [139,140] or in vitro using display technologies [141]. The emergence of novel in silico antibody discovery methods have allowed the generation of antibodies in a cost- and time-effective manner [142]. AbDesign, a structure-based algorithm that utilizes combinatorial and energy-based design, has been successful in designing antibody fragments that target insulin and mycobacterial acyl-carrier protein [143]. Similar successes have been made using structure-based procedures such as OptMAVEn [144] and Rosetta antibody design [145]. More recently, the Cascade method [128] has been developed to design complementary peptides to a target epitope that can be grafted to the complementarity-determining region of an antibody scaffold. The Cascade method has previously been employed in designing conformation-specific antibodies to target Aβ42 amyloids and oligomers [127,146].

### 4.4. PET Neuroimaging of Amyloids in the Brain

PET imaging utilizes radiotracers to screen the presence of amyloids in the brain in vivo [147]. The radiotracers are characterized by high lipophilicity, which allows them to pass across the Blood-Brain-Barrier. Pittsburgh compound B (^11^C-PiB) is a fluorescent analog of ThT and was first used to trace amyloid fibrils using PET in 2004. ^11^C-PiB is currently among the most used radio ligands for PET imaging of cerebral Aβ pathology [148] (Table 2). Amyloid-PET can distinguish dementia from mild cognitive impairment and normal aging. However, it cannot discriminate different forms of dementia. Furthermore, these techniques are expensive and current clinically used radiotracers only reflect the level of insoluble aggregates in the brain. In contrast, novel antibody-based radioligands are under development to detect soluble oligomeric and protofibrillar Aβ forms and have been tested in transgenic mice [149,150].

### 4.5. Combined Detection

Neurodegenerative diseases share many symptoms and pathological mechanisms. In particular, aggregates of the same amyloidogenic protein can exist in people affected by different forms of neurodegeneration [151]. For example, Aβ and p-tau aggregates can also be found in patients with DLB, and α-syn aggregates in patients with AD. The co-occurrence of Aβ, α-syn, and tau suggests an overlap between AD, tauopathies and synucleinopathies [152]. Amyloid aggregates can also be present in people who do not show any disease symptoms [151]. These observations lead to the conclusion that, in some cases, monitoring one individual amyloidogenic protein alone may not be sufficient for delivering an accurate diagnosis.

Various studies have assessed the diagnostic performance of monitoring the levels of different amyloidogenic proteins together (Table 2). For example, the quantification of the CSF concentrations of both p-tau181, carrying the phosphorylated Thr181, and the ratio Aβ42/Aβ38 was able to differentiate AD from other neurodegenerative diseases [66,104]. A recent study conducted on a cohort of 4444 participants over a period of 14 years was able to associate the plasma levels of NFTs and Aβ42 with the risk of developing AD and all-cause dementia [153]. Furthermore, the analysis of the CSF levels of both total α-syn and total tau could help the identification of synucleinopathies over other neurodegenerative diseases [68]. A recent work has shown that the determination of the ratio of oligomeric-α-syn/total α-syn, p-α-syn129, and p-tau181 in the CSF was able to identify PD patients from controls [154]. It has also been shown that an increase in glycation, α-syn Tyr 39 nitration and pTyr125, and a decrease in SUMO-1 levels in blood samples was associate with PD [117]. Finally, it has been found that the concentration in the CSF of total TDP-43 and the ratio total tau/pThr181-tau discriminate ALS/FTD patients from healthy controls [155]. In Table 2, we summarize the main proteins which can be monitored in combination for diagnostic purposes.

**Table 2 ijms-22-04128-t002:** Main current and potential diagnostic markers of neurodegenerative diseases with main references.

Type of Diagnostic Markers			Key Remarks	Refs.
**Genetic**				
Protein	Gene	Mutations	Pathological implications	
APP	*APP*	E693Q, A692G	Early onset familial AD	[35,36]
α-Syn	*SNCA*	A53T, G51D, H50Q, E46K, A30P, locus amplification	Early onset familial PD	[46,47,48,156]
TDP-43	*TARDBP*	G294A, Q331K, M337V, K181E	Sporadic and familial ALS	[58,59]
FUS	*FUS*	R521C, R521H	Early onset ALS	[63]
**Neuroimaging**				
Protein species			Analytical technique	
Aβ and tau aggregates			*PET with ^18^F-FDDNP*. Lacks protein specific and is unable to distinguish different forms of neurodegeneration.	[147]
Tau aggregates			*PET with ^18^F-AV1451 or ^18^F-GTP1*. High affinity for tau aggregates. Not able to distinguish different forms of tauopathies.	[147]
Aβ aggregates			*PET with ^11^C-PiB and Florbetapir (^18^F)*. High affinity for Aβ plaques. Not able to distinguish different forms of neurodegeneration.	[147,148]
**Biomarkers**				
Protein variants			Analytical techniques	
CSF Aβ42			*Immunoassays*. In the majority of cases, it is present at lower concentration in patients affected by AD and other forms of dementia. It is not accurate in distinguishing different forms of neurodegeneration.	[78]
CSF Aβ42/Aβ40 ratio			*Immunoassays*. Slightly better diagnostic performance for AD than Aβ42 alone. Usually, it is used in combination with other potential biomarkers (e.g., tau).	[78,103]
CSF Aβ42/Aβ38 ratio			*Immunoassays*. Better diagnostic performance for AD than Aβ42 alone. Also used in combination with other potential biomarkers (e.g., p-tau).	[78,103,107]
CSF total tau			*Immunoassays*. Generally present at higher concentrations in the CSF of AD patients. It is not disease-specific when used alone and usually measured in association with other amyloidogenic proteins (e.g., Aβ42).	[67]
CSF p-tau217, p-tau181			*Immunoassays and MS*. Their concentrations correlate with AD. P-tau181 is generally used in combination with other potential biomarkers, including total tau and Aβ42, or with the Aβ42/Aβ38 ratio.	[107]
Plasma p-tau217			*Immunoassays and MS*. Able to discriminate AD from other forms of neurodegeneration.	[86]
CSF and plasma p-α-syn129			*Immunoassays*. Good diagnostic performance for PD when used in combination with other potential biomarkers, including tau and α-syn oligomers.	[109,110,154]
CSF Aβ oligomers and α-syn oligomers			*Immunoassays and single-molecule approaches*. Good potential diagnostic performance for AD or PD, respectively. However, their use as biomarkers is not widely implemented yet. Some works have used them in combination with other potential biomarkers.	[109,132,154]

## 5. Recent Advances in Detection Technology

Given their affordable costs and the possibility of easily making them high-throughput, immunoassays, such as ELISA and immunoblotting, have been the most commonly used techniques to quantify amyloidogenic proteins’ concentrations in body fluids and tissues. Other approaches are also in use, including PET, MS, and microscopy, and novel detection technologies with ultra-sensitivity are emerging.

An important example is provided by MS. A recently established capillary isotachophoresis–electrospray ionization MS could detect picomolar concentrations of Aβ [157]. In another study, an automated clinical mass spectrometer could detect different Aβ variants in the CSF in a multiplex manner [158]. Noteworthy, both approaches were antibody-free and did not require an immuno-enrichment step. Aβ42/Aβ40 ratios in CSF could also be determined by LC-MS/MS assay with a high clinical sensitivity [159]. The benefits of MS include small sample size, fast turnaround time, broad applicability and sensitivity.

Single-particle analysis of amyloids can be performed by microscopic methods including fluorescence, atomic force, and electron microscopy. Notably, fluorescence-based methods provide ultrasensitive detection of individual amyloid fibrils and oligomers in neurodegenerative diseases. Furthermore, super-resolution methods offer insight into structural properties and surface hydrophobicity [17].

Nanopore sensing is a non-optical technique that has recently been demonstrated to allow single-molecule analysis of polymeric proteins and could be extended to amyloids and oligomers [160,161]. In nanopore sensing, a biomolecule is translocated through a nanopore embedded within a thin dielectric membrane separating two chambers with electrolytes. Distinct conformations of the biomolecule can be characterized upon its translocation through the nanopore by the analysis of the change in the ionic conductance of the pore [162,163].

Other recent methods include amyloid seeding assays, such as protein misfolding cyclic amplification [164], and real-time quaking-induced conversion (RT-QuIC) [165]. In these assays, the composition and number of amyloids in biological samples is determined by the ability of these samples to induce the aggregation of a recombinant monomeric amyloidogenic protein using ThT-based aggregation measurements. These assays were established for several amyloidogenic proteins. Especially, RT-QuIC assay for α-syn showed high diagnostic sensitivity for PD and DLB [165].

Both MS and microscopic techniques require sophisticated instrumentation. ELISA kits for several biomarkers including Aβ42, p-tau, total-tau and α-syn are commercially available. However, ELISA can be labor-intensive to set up and with confined sensitivity. Various high sensitivity ELISA techniques have been developed. For instance, researchers have measured total α-syn concentration in body fluids using immobilized lipids [166]. Also, ELISA has been coupled with other detection technologies, such as novel plate-based electrochemiluminescence [167]. This approach achieved markedly shortened processing time, with smaller sample volume requirements and simultaneous processing of multiple biomarkers [167]. Besides electrochemiluminescence, digital ELISA has also been developed with single molecule array (Simoa) technology. This method is reported to have increased sensitivity for Aβ42 detection in the human plasma (in the pM range) [168]. Moreover, a surface-based fluorescence intensity distribution analysis (sFIDA) assay was established resembling a sandwich ELISA where Aβ oligomers were immobilized on the functionalized glass surface via antibodies, imaged by high-resolution fluorescence microscopy [169].

The Multi-Analyte Profiling (xMAP) platform stands out from a wide range of approaches based on its multiplexing capability. Simultaneous quantification of up to 100 samples in a single assay could be achieved on a semi-automated assay. Studies provided that xMAP data for total tau, p-tau, Aβ40 and Aβ42 correlated well with research-based ELISA values with higher sensitivity and specificity [170].

Immuno-polymerase chain reaction (I-PCR) utilizes real-time PCR (also known as quantitative PCR) to combine nucleic acid amplification with antibody-based assays to increase 10 to 109-fold sensitivity of conventional immune assays. Researchers quantified multiple phosphorylated tau epitopes using I-PCR in CSF [171] and developed a nano-I-PCR approach which utilized gold nanoparticles functionalized with a tau-specific monoclonal antibody for total tau quantification in CSF samples [172]. The level of total Aβ40 present in microdissected neurons could also be quantitated using I-PCR with high sensitivity and detection range [173]. This technique is suitable for small sample volumes, provides rapid time to results and can be amenable to multiplexing.

Point-of-care (POC) diagnosis is, undoubtedly, an emerging trend. This approach allows conventional ELISA, Luminex xMAP and qPCR to be developed into inexpensive, portable and easy-to-use POC devices [174]. Paper-based ELISA is the simplest option for POC diagnostics.

## 6. Conclusions and Potential Future Directions

Disease biomarkers represent an essential requirement for the development of accurate diagnostic approaches. In this review, we have discussed how amyloidogenic proteins hold potential as biomarkers for neurodegenerative diseases and described detection technologies to assess their concentrations in the body (Figure 3). Besides genetic mutations, many PTMs and specific conformations of amyloidogenic proteins are associated with disease and are emerging as potential biomarkers (Table 1 and Table 2). In this context, noteworthy among all aggregated conformations are the oligomers, which are highly toxic and regarded as major players in the disease onset and progression.

The use of amyloidogenic proteins as biomarkers comes with challenges. Firstly, neurodegenerative diseases share some key pathological mechanisms, including the formation of aggregates by the same amyloidogenic protein (e.g., Aβ and p-tau deposits can be found in patients with DLB) [152]. This makes it difficult to distinguish one form of neurodegeneration from another based on the detection on one specific amyloidogenic protein alone. Furthermore, amyloidogenic proteins are difficult to access within the CNS and their concentrations in accessible body fluids have noticeable fluctuations, particularly at early stages of disease.

In our opinion, promising diagnostic strategies that may overcome these issues are those based on the detection of multiple amyloidogenic proteins or protein features (Section 4.5). Additionally, amyloidogenic proteins could be monitored in conjunction with other types of biomarkers, such as metabolites. Recent investigations have shown that metabolic pathways are affected in neurodegeneration, and detection platforms have been developed for metabolic profiling [175]. For examples, nuclear magnetic resonance and MS have been successfully employed to determine metabolic changes in cellular systems [176], post-mortem brain samples [177], and CSF from patients [178]. Combined approaches could also involve neuropsychological assessment and neuroimaging.

Furthermore, amyloidogenic proteins can also be detected in regions of the body besides the CNS and body fluids. For example, aggregated forms of α-syn have been found in the digestive system of PD patients [179]. The diagnostic relevance of this finding is twofold: it shows that other parts of the body can be examined for potential biomarkers of neurodegeneration [180]; it also implies that patients may show symptoms/disorders that are apparently unrelated with the neurodegenerative condition but instead could be used for early diagnosis.

Sensitivity and specificity are important attributes of detection technologies for amyloidogenic proteins. In Section 4.3 and Section 5, we described several promising approaches which are currently under development. These are based on biosensors, single-molecule detection, and molecular probes (e.g., antibodies). In our view, antibody-based approaches in particular hold a great potential as they allow detection in complex mixtures. Moreover, antibodies can be developed to target different protein features, including PTMs and conformations (e.g., the oligomers).

In conclusion, amyloidogenic proteins are appealing potential biomarkers of neurodegeneration. Their diagnostic success is intertwined with the development of combined detection strategies, involving other types of biomarkers, organ systems, and ultra-sensitive technologies.

## Figures and Tables

**Figure 1 ijms-22-04128-f001:**
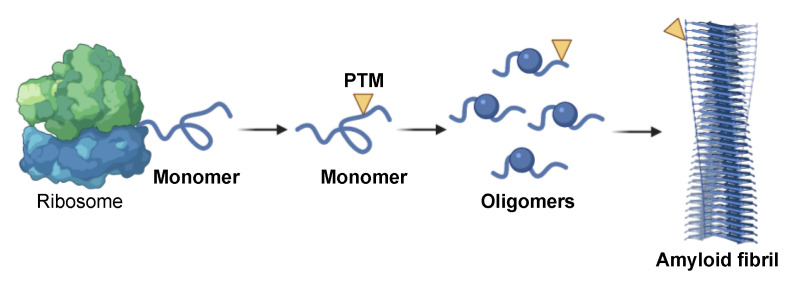
Schematic representation of amyloid aggregation. In normal conditions, proteins are newly synthesized on ribosomes and released to play biological functions. In stress conditions or as a consequence of pathological post-translational modifications (PTMs), these proteins can assemble into small soluble oligomers and ultimately form insoluble amyloid fibrils. PTMs may occur at any state of this process, affecting the formation, stability and toxicity of these aggregates.

**Figure 2 ijms-22-04128-f002:**
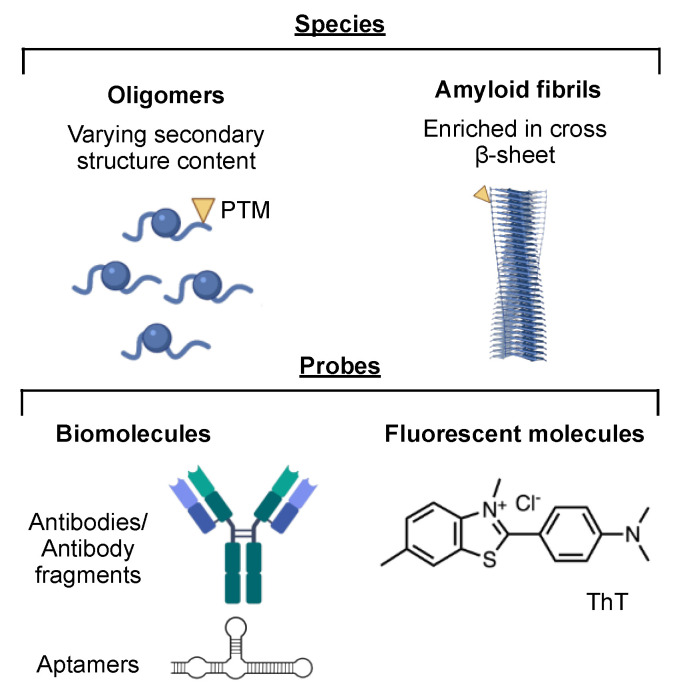
Aggregated protein species and representative probes used for their detection.

**Figure 3 ijms-22-04128-f003:**
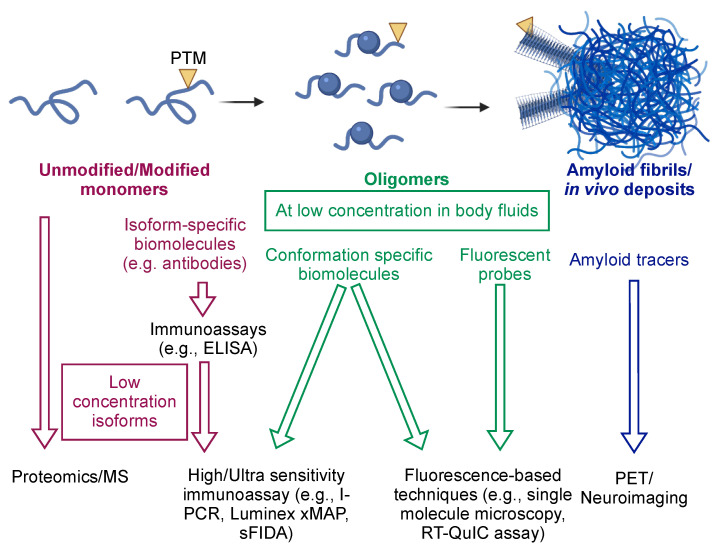
Detection of amyloidogenic proteins at different stages of protein aggregation. Main representative strategies and techniques are shown.

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
