# Peer review of "The Diagnostic Potential of Amyloidogenic Proteins"

_ijms, 2021, doi:10.3390/ijms22084128_

Round 1

Reviewer 1 Report

The paper presented by Jin et al. is a review aimed to describe the latest advances in diagnostics field of amyloid neurodegenerative diseases. These advances include the use of  specific biomarkers and innovative diagnostic techniques. The review focuses on the proteins involved in the most severe conditions for human health, like amyloid-beta (Aß) and tau for AD, ⍺-synuclein (⍺-syn) for PD, Fused in Sarcoma (FUS) and the TAR DNA-binding protein 43 (TDP-43) for ALS and FTD. Also the post-translational modifications are discussed. 

The review is well written and I believe it can be very useful for understanding the general picture of diagnostic tools, that, together with therapies, are crucial for preventing amyloid  diseases development.

I think that the paper could be even more complete if information on NMR metabolomics approach used for probing the metabolic changes, for example in cerebrospinal fluid, associated to amyloid were also described from a diagnostic point of view.

Author Response

We thank the reviewer for their positive comments. Following their suggestion, we have included information regarding NMR metabolomics in the Conclusions section (Section 6).

Reviewer 2 Report

Following the analysis of the manuscript titled "The Promise of Amyloid-Based Diagnostics", I appreciate the article's topic is interesting, the presentation of the information is clear and properly structured, the figures are expressive and reinforce the presented notions, and the manuscript is well documented.

Please clarify the title of the manuscript, as the one from the pdf is different from the one uploaded in the submission process.

Please refer to figure 2 and figure 3 in the text.

Please outline one or more clear conclusions in a separate section.

Author Response

We thank the reviewer for their positive feedback on our manuscript.

-We have now clarified the title of the review, which is The Diagnostic Potential of Amyloidogenic Proteins

-We have included references to Figures 2 and 3 in the text

-We have revised the Conclusions section (Section 6), which now focuses on the main points of the review.

Reviewer 3 Report

This narrative review describes the use of amyloidogenic proteins as a biomarker. The article is interesting, it addresses a current topic. In my opinion, this work is acceptable with few minor changes.

  • The title and abstract reflect the content of the work.
  • The manuscript is concisely written, and the conclusions drawn are supported by the data and the adequate referencing of past studies.
  • The study is scientifically sound and advances further the knowledge in this very important area of neuropathy.
  • Please add tables of all potential markers including genetic markers and add their specificity and accuracy. Also, add their methods of the assay.

Author Response

We thank the reviewer for their positive feedback. We have added a new table (Table 2) which reports information regarding the current most investigated potential markers.